# Imitation Game: Toward Comprehensive Evaluation on Personalized Role-Playing on Social Media

## Abstract

Social simulation observes the emergence of complex macro-scopic social patterns from individual interactions within a virtual social environment. A common practice in social simulation is to model individual as a statistical average of a specific group, failing to capture individual heterogeneity. To model such heterogeneity, we propose a personalized role-playing task in the context of social media, which provides environment for social simulation with massive authentic user interactions. As no public social media dataset concentrates especially on historical interactions of individual user for personalization, we collect data from reddit and construct our own dataset, consisting of 67 users, 7K posts, and 21K comments. And we introduce three key dimensions for personalized role-playing and conduct comprehensive evaluation on feasible role-playing methods. The results yield the following key findings:(1) existing methods struggle to achieve fine-grained personalized modeling; (2) merely scaling model parameters or applying reasoning models is insufficient to substantially enhance the level of personalization; (3) the evaluated methods exhibit significant vulnerability to noise within interaction context.

## 1 Introduction

Large Language Models (LLMs) have made great progress in long-context comprehension (Zhao et al., 2024a; Qiu et al., 2025), emotion perception (Chen et al., 2023; Zhao et al., 2024b) and complex reasoning (Juneja et al., 2024; Cai et al., 2025), promoting their application in social simulation. In LLM-based social simulation, LLMs are usually endowed with various social identities and characteristics to simulate individual reactions in specific scenarios. Precise simulation at the individual level is fundamental to the credibility and effectiveness of social simulation. As a result, LLM role-playing, which aims for imitation with high-fidelity, is widely integrated into social simulation. However, existing LLM role-playing (Shao et al., 2023; Wu et al., 2024) excels at simulating archetypal figures, such as artistic characters and celebrities, as these figures typically possess detailed profiles and abundant interaction records. However, its ability to simulate ordinary individuals given limited observation remains to be explored. Since ordinary people's behaviors are most vividly reflected in social media interactions, it is very practical to investigate LLM role-playing in such contexts.

Role-playing for social media users is non-trivial, as it involves three main challenges: 1) **incomplete or missing profile**: The majority of social media users refrain from uploading or displaying genuine personal information for privacy concerns, rendering profile-driven role-playing methods inapplicable; 2) **interaction data sparsity**: social media users tend to comment on a small number of posts that interest them, and a significant proportion of social media users are passive observers, rarely contributing to discussions; 3) **language dependency**: unlike artistic works where characters can be portrayed through facial expressions and body language, accessible behavior of social media user primarily consists of posts and comments in natural language form, which are typically concise, emotional, and informal, making language modeling highly challenging.

In this paper, we propose a novel task called **P**ersonalized **R**ole-playing on **S**ocial **M**edia (PRISM). The task requires LLMs to simulate ordinary social media users, mimicking their perspectives,

stances, and linguistic habits to participate in discussions on specific topics. The task is designed in utterance level, where a simulated comment is compared with the corresponding ground-truth comment from target user for evaluation. To study this new task, we collect the most recent user data and construct a high-quality benchmark called IDRole, since no public social media dataset focuses especially on historical interactions of individual user for personalization and data previously collected from social media platform poses a potential risk of data contamination. We first select topic communities with high popularity such as science and technology. Then we scan the most popular posts within each community and identify active users from the posters and commenters of these posts. After retrieving the historical posts and comments from each active users, we refine the raw data and reorganize each sample into a free-form text completion problem with a given context. Finally we obtain IDRole, containing 67 users and 21K samples refined from 7K posts and 21K comments. Each sample consists of an original post, the conversational context, and a ground-truth comment of the target user.

To promote further development, we conduct a comprehensive evaluation and analysis of role-playing methods on IDRole, including few-shot prompting, SFT, DPO (Rafailov et al., 2023), and GRPO (Shao et al., 2024). Our evaluation framework for role-playing methods integrates both classical similarity and LLM-as-Judge metrics that consider three key personalization dimensions. We assess the performance of these methods on both open-source and commercial LLMs. Additionally, we explore the potential of small language models in personalized role-playing, analyze the correlation between personalized role-playing performance and length of target ground-truth comments, and study the robustness of role-playing methods against noise injected into discussion context. The key findings derived from the experimental results are as follows: (1) all evaluated role-playing methods yield unsatisfactory performance in personalized role-playing on social media; (2) merely scaling model parameters or applying reasoning models is insufficient to substantially enhance the level of personalization; (3) the evaluated role-playing methods exhibit vulnerability to the noise in the context.

Overall, the main contributions of this work can be summarized as follows:

- We propose the personalized role-playing task on social media. To the best of our knowledge, we are the first to investigate the ability of role-playing methods to model general users from a generative perspective, based on real-world social media data.

- We construct a new benchmark called IDRole, considering the lack of readily available datasets for task evaluation and potential data contamination issue. IDRole contains 67 social media users and 21K samples at utterance-level refined from 7K posts and 21K comments.

- We conduct a comprehensive evaluation and analysis on role-playing methods. The experiment results demonstrate that existing role-playing methods struggle to perform well on IDRole and highlight the need to develop strong and robust personalized role-playing method on social media with sufficient attention to the potential of small language models.

## 2 RELATED WORK

### 2.1 SOCIAL SIMULATION

Social simulation is providing new tools and perspectives for sociological research. Based on simulation granularity, social simulation can be categorized into the simulation of individual behavior, local interactions, and society system.

Individual simulation utilizes LLM agents to simulate specific individuals, focusing on modeling the characteristics of a single person. In individual simulation, explicit characteristics of individuals typically derive from demographic information or known character-related knowledge, while implicit characteristics require mining from behavioral and psychological activities. Horton (2023) utilizes LLM to simulate individual behavior in a predefined economic scenario, which is assigned different social preferences such as fairness, total benefit, and personal benefit. Argyle et al. (2023) construct GPT-3 simulated samples matched to the demographic characteristics of participants in the American National Election Studies and instruct these samples to simulate human voting choices. Ge et al. (2024) incorporate character-specific attributes into the data synthesis prompt, which steer the LLM to generate unique synthetic data aligning with the designated persona's perspective.

Simulation for local interactions involves multiple agents with designated roles within specific scenarios, focusing on both agent-agent and agent-environment interactions, with simulation processes driven by character motivations or predefined tasks. Zhou et al. (2023) construct a social interaction environment featuring diverse social scenarios, where agents attempt to achieve social goals such as cooperation and competition through various forms of communication. Qian et al. (2024) simulate the software development process, wherein software agents, embodying professional roles such as programmers, code reviewers, and test engineers, engage in collaborative dialogue to accomplish tasks specific to each development phase.

Social simulation constitutes a significant extension of local simulation in both spatiotemporal scale and system complexity, aiming to capture and reproduce emergent macroscopic social dynamics from microscopic interactions over a broader scope. Mou et al. (2024) simulate the reaction of group to specific social movements on social media, with ordinary users modeled by mathematical Agent-based models (ABMs). Zhang et al. (2025) construct a world model for social simulation containing 10 million individuals and personalize each user with predicted demographic attributes. During evaluation, the macro-level metrics are computed by aggregating individual questionnaire responses.

While existing studies in social simulation have demonstrated considerable efficacy, they typically parameterize individuals using a limited set of predefined attributes. Consequently, there is a gap in research on the fine-grained simulation of individuals within more authentic and interactive environments.

## 2.2 LLMs-based Role-Playing

The advancements in comprehension and generative capabilities of LLMs have laid the foundation for role-playing. Existing role-playing methods can be categorized into nonparametric prompt engineering and parametric fine-tuning. The subjects of role-playing are typically real celebrities or fictional characters from literature and art.

In nonparametric prompt engineering, Xu et al. (2024) leverage character descriptions and memory retrieval enabling general LLMs to make persona-driven decisions. In parametric fine-tuning, Wang et al. (2024) constructed 100 character profiles from public scripts, subsequently generating knowledge-infused question-answering pairs with GPT. Li et al. (2023) further broadened the scope of character sources to novels, TV shows, and wiki, while enriching multi-turn dialogue data by directly extracting and synthesizing. Zhou et al. (2024) developed Chinese role-playing models allowing flexible configuration in attributes and styles of characters. Shao et al. (2023) extracted and refined characters' experiences with the assistance of GPT according to profiles collected from wiki. Lu et al. (2024) proposed a self-alignment approach and construct fine-tuning data based on the responses of target LLMs to role-specific and out-of-scope queries.

Although existing work has made much progress, the profile of characters are often detailed and the behavioral data used for modeling is generally abundant. Role-playing for general users on social media with sparse data remains unexplored.

## 3 TASK DEFINITION

Conventional role-playing tasks often establish a predetermined scenario $S$ where each agent $a_i$ is assigned detailed character profile $p_i$, typically encompassing attributes such as identity, background, objective, and personality. Agents need to interact with each other guided by their goals to advance the plot. In a complete interaction cycle, each agent's response conditioned on profile $p_i$ is shaped by the scenario and the preceding communicative acts of other agents, which can be formalized as:

$$o_i = G_i(p_i, S, o_1, o_2, ..., o_{i-1}), i \in \{1, 2, ..., N\}, \tag{1}$$

where $G_i$ and $o_i$ represent the generating function and the response of role $r_i$, respectively. The sampled responses are subsequently used to evaluate the character's fidelity.

Considering the forms of user activity in social media contexts, modifications to the above task are required to ensure its suitability. Users on social media post and comment on specific events and

topics. User comments, as a form of personalized expression, can be used to achieve and evaluate personalized role-playing. Let $P$ be a root post under which the target user $u_o$ leaves a comment $c_0$. $P$ comprises a title $t$ and a body $b$. We define the interaction history $H$ preceding $c_0$ as $\{(u_i, c_i)\}_{i=1}^N$, where $c_i$ is the $i$-th comment created by user $u_i$ below the root post. We task LLMs to generate a simulated comment $\hat{c}_0$ in the context of $P$ and $H$:

$$\hat{c}_0 = G(P, H), \tag{2}$$

$$H = [(u_1, c_1), (u_2, c_2), ..., (u_N, c_N)] \tag{3}$$

Ideally, the personalized role-playing system would grasp the distinct linguistic patterns and semantic nuances within the target user's comments.

## 4 BENCHMARK CONSTRUCTION

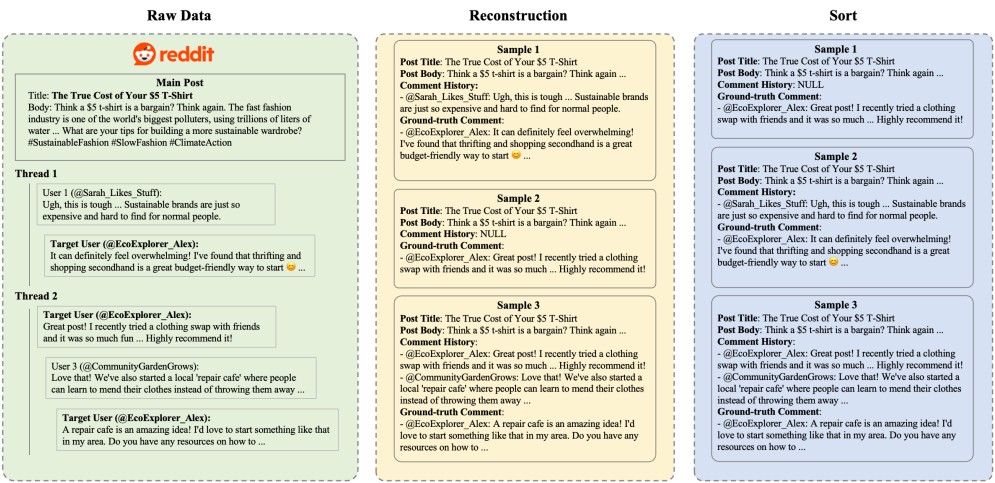

Figure 1: An overview of IDRole construction. The raw data from reddit consists of main posts and multiple comment threads. We reconstruct the raw data into samples containing post information, comment history, and ground-truth comments. The samples are sorted by the depth of the target comments to avoid possible data leakage.

Considering personalized role-playing should be applicable to diverse target individuals, we collect user data from a diverse range of interest-based communities instead of data of notable users. The overview of IDRole construction is shown in Figure 5. We first select Reddit as data source since it contains millions of communities of interest called subreddits and provides free and convenient data access API. Then we obtain target users from selected influential subreddits such as science and technology. Specifically, we scan the most recent top hot posts in each subreddit and record the posters and commenters within the first two levels in comment thread. Active users are filtered out according to the influence of posts and comments measured by karma value from Reddit. The scan will terminate automatically once the number of active users reaches the preset threshold of 80. For each active user, we request the API and retrieve the most recently delivered posts and comments up to the API's retrieval limit chronologically. We manually inspect the retrieved posts and comments, excluding active users who had set the content to private and finally obtaining 67 valid active users. Subsequently we reconstruct the entire comment thread for each retrieved comment by recursively traversing upwards according to the parent pointer of each comment node. The traversal enables us to gather all comments preceding the one of target user and the root post, along with their respective authors. We totally obtain 27,585 raw posts and 21,399 raw comments with their context.

In processing the raw posts, we filter out posts containing multi-modal content including images and videos and finally obtain 7,221 text-based posts. In processing the raw comments with their context, we assign a "NULL" string to the comment history, post title, and post body fields when they are found to be empty strings, obtaining the same number of processed comments. To adapt to the personalized role-playing task on social media, we treat the combination of a main post, comment

history, and a comment delivered by target users as a single sample. To avoid possible data leakage, where the comment history of one sample might contain the target comment of another, we sort the samples by the depth of the target comment within the comment thread in ascending order. Samples with greater target comment depth are preferentially selected for the test set.

# 5 EXPERIMENTS

## 5.1 MODEL AND DATA PREPARATION

We conduct personalized role-playing evaluation based on LLaMA3-8B-Instruct and Qwen2.5-7B-Instruct. In zero-shot setting, we also report the performance of reasoning model Qwen3-8B fine-tuned on Chain-of-Thought(CoT) data of DeepSeek-R1, LLaMA3-70B-Instruct and closed-source commercial model GPT-4.1. To analysis the potential of small language models in personalized role-playing, we also conduct evaluation on LLaMA-3.2-3B-Instruct and Qwen2.5-3B-Instruct. For data preparation, we sort the processed samples by the depth of target comment within comment thread to avoid data contamination. We then split the dataset, with the first 60% of the samples as the training set and the remaining part as the test set. GPT-4.1 is employed for LLM-as-Judge. The prompt templates for model inference and LLM-as-Judge are provided in Appendix.

## 5.2 EVALUATION

### 5.2.1 METRICS

We evaluate the quality of personalized role-playing from the prospective of generation. Given a simulated comment and the corresponding golden comment from the target user, we calculate the lexical and semantic similarity between the two above comments based on ROUGE score and BERTScore. To further evaluate the personalization of simulated comments, we introduce LLM-as-Judge and summarize three key dimensions, which are detailed as follows:

- Semantic & Stance Fidelity (SS): Simulated comments are expected to not only preserve the core meaning of the golden comments, but also to faithfully reproduce the authors' attitude and stance.

- Contextual & Interactional Coherence (CI): Simulated comments are expected to be thematically relevant and engage proactively with the preceding comments within the comment thread.

- Linguistic & Stylistic Fidelity (LS): Simulated comments are expected to adopt the similar linguistic features such as grammatical structure and lexicon, while imitating the specific style and tone.

### 5.2.2 EVALUATED METHODS

We evaluate four methods applicable to personalized role-playing including few-shot prompt, SFT, DPO, and GRPO from the technical perspective of prompt engineering, fine-tuning, and reinforcement learning.

- Few-shot prompt (FS): Considering the scarcity of user data, we first evaluate LLMs under few-shot setting. Specifically, we select two comments with their context for each target user as demonstrations. LLMs are then prompted to infer persona, learn expressive patterns, and output personalized comments.

- Supervised Fine-Tuning (SFT): SFT is typically used to improve the task-solving and instruction-following capabilities of LLMs. We conduct SFT training on golden comments conditioned on their context. The conciseness and stylistic diversity inherent in social media comments motivate an investigation into the effectiveness of SFT within this challenging, data-limited domain.

- Direct Preference Optimization (DPO): DPO derives the analytical expression for the optimal reward function and constructs a loss function depending solely on the current policy, the reference policy and preference pair for direct optimization of model parameters, eliminating the need to train a reward model. We utilize DPO to model personalization from the nuances between golden comments of target users and general comments produced by general LLMs.

- Group Relative Policy Optimization (GRPO): GRPO samples a set of completions for each prompt and calculate the relative advantage of each completion within the group according to its reward. The loss function of GRPO encourages model to maximize the advantage while controlling the KL divergence between current and reference policy. We utilize Qwen2.5-32B-Instruct to score the completion and return rewards averaged from three LLM-as-Judge metrics for model update.

The implementation details for the evaluated methods are provided in Appendix.

## 5.3 EXPERIMENTAL RESULTS AND ANALYSIS

### 5.3.1 OVERALL RESULTS

Table 1: The average results across all target users for each evaluated role-playing method. In each group characterized by the original LLM, **bold** figures indicate the best result in each evaluation dimension, while the underlined figures denote the second best one.

| Method | BERTScore | ROUGE | | | LLM-as-Judge | | | |
|---|---|---|---|---|---|---|---|---|
| | | ROUGE-1 | ROUGE-2 | ROUGE-L | SS | CI | LS | Avg. |
| LLaMA-3.2-3B-Instruct | | | | | | | | |
| Few-shot | 0.836 | 0.124 | 0.011 | 0.091 | 1.404 | 1.998 | 1.555 | 1.652 |
| RAG | 0.832 | 0.143 | **0.018** | **0.099** | 1.529 | 2.112 | 1.640 | 1.760 |
| SFT | **0.837** | 0.117 | 0.013 | 0.088 | 1.309 | 1.828 | 1.441 | 1.526 |
| DPO | 0.833 | 0.095 | 0.010 | 0.075 | 1.310 | 1.825 | 1.410 | 1.515 |
| GRPO | 0.825 | **0.157** | **0.018** | 0.096 | **1.925** | **2.814** | **1.998** | **2.246** |
| LLaMA-3-8B-Instruct | | | | | | | | |
| Few-shot | 0.828 | **0.155** | 0.015 | 0.096 | 1.863 | 2.764 | 1.906 | 2.178 |
| RAG | 0.834 | 0.147 | **0.020** | **0.102** | 1.720 | 2.501 | 1.851 | 2.024 |
| SFT | **0.838** | 0.113 | 0.013 | 0.087 | 1.361 | 1.974 | 1.473 | 1.603 |
| DPO | 0.830 | 0.098 | 0.010 | 0.074 | 1.355 | 1.951 | 1.485 | 1.597 |
| GRPO | 0.831 | 0.152 | 0.014 | 0.097 | **2.076** | **3.166** | **2.164** | **2.469** |
| Qwen2.5-3B-Instruct | | | | | | | | |
| Few-shot | 0.834 | 0.132 | 0.012 | 0.088 | 1.821 | 2.672 | 1.875 | 2.123 |
| RAG | 0.833 | 0.147 | **0.020** | **0.097** | 1.840 | 2.637 | 1.859 | 2.112 |
| SFT | **0.838** | 0.118 | 0.014 | 0.088 | 1.434 | 2.044 | 1.579 | 1.686 |
| DPO | 0.837 | 0.112 | 0.011 | 0.082 | 1.542 | 2.258 | 1.651 | 1.817 |
| GRPO | 0.828 | **0.153** | 0.016 | 0.092 | **1.980** | **2.963** | **2.013** | **2.319** |
| Qwen2.5-7B-Instruct | | | | | | | | |
| Few-shot | 0.834 | 0.127 | 0.010 | 0.086 | 1.782 | 2.622 | 1.844 | 2.083 |
| RAG | 0.834 | 0.144 | **0.016** | 0.095 | 1.920 | 2.817 | 1.958 | 2.232 |
| SFT | **0.837** | 0.139 | 0.013 | **0.096** | 1.709 | 2.528 | 1.856 | 2.031 |
| DPO | 0.833 | 0.097 | 0.007 | 0.073 | 1.668 | 2.455 | 1.779 | 1.967 |
| GRPO | 0.826 | **0.147** | 0.014 | 0.090 | **2.163** | **3.212** | **2.171** | **2.515** |
| Qwen3-8B | | | | | | | | |
| Few-shot | 0.829 | 0.137 | 0.012 | 0.089 | 1.782 | 2.658 | 1.906 | 2.115 |
| LLaMA-3-70B-Instruct | | | | | | | | |
| Few-shot | 0.831 | 0.152 | 0.014 | 0.097 | 2.076 | 3.166 | 2.164 | 2.469 |
| GPT-4.1 | | | | | | | | |
| Few-shot | 0.833 | 0.149 | 0.014 | 0.094 | 2.456 | 3.643 | 2.602 | 2.900 |

The average results across all target users for each evaluated role-playing method are shown in Table 1. We note the following key observations throughout our experiments:

- All evaluated methods do not perform well in personalization for the group of target users: On BERTScore, the performance of the methods varies slightly, indicating that no significant semantic distortion occurs on the validation set. On ROUGE scores, the generally low values show that LLMs adhere to their own generative patterns rather than emulating the lexical habits of specific

users. According to the scores from LLM judge, maintaining semantic and stance fidelity, along with linguistic and stylistic fidelity, pose more challenges for LLM role-playing than achieving context and interaction coherence, which is supported by the inherent abilities of base LLMs to understand and generate text.

- GRPO consistently outperforms other methods across multiple evaluation metrics, and RAG performs relatively better in prompt-based methods. We can observe that performance differences between methods are primarily reflected in the LLM-as-Judge metrics. The SS, CI, and LS scores of GRPO are consistently higher than other methods, demonstrating that the point-wise personalized rewards provided by an external reward model that compares simulated and ground-truth comments can effectively improve policy model's personalization capabilities within a reinforcement learning framework. The performance degradation of other training-based methods may stem from the scarcity of user data and the discrepancies between individual values of users and alignment values of LLMs. RAG may introduce inconsistencies in performance by providing contextual knowledge that conflicts with the model's internal parameter knowledge. Nevertheless, the ROUGE scores and LLM-as-Judge scores demonstrate its effectiveness in improving the performance of personalization.

- Merely scaling model parameters or applying reasoning models is insufficient to substantially enhance the level of personalization: To investigate the effect of scaling model parameters, we conduct evaluation on LLaMA-3-70B-Instruct and compare the results with LLaMA-3-8B-Instruct. We can observe that there is almost no improvement in BERTScore and ROUGE scores. For LLM-as-Judge, scaling up model parameters has a more pronounced effect on the CI score than on the SS and LS scores, illustrating that merely scaling up models is insufficient to substantially promote the level of personalization. We also explore the performance of reasoning model under zero-shot setting. Compared with LLaMA-3-8B-Instruct and Qwen2.5-7B-Instruct, Qwen3-8B finetued with Chain-of-Thought data distilled from DeepSeek-R1-0528 does not exhibit performance advantages considering the inference-time computing, with only linguistic style fidelity score achieves the best among them.

### 5.3.2 ANALYSIS OF SMALL LANGUAGE MODELS

Small language models feature in low latency, low computational costs, and on-device deployment, exhibiting potential in large-scale social simulation, extensive character customization, and privacy-concerned scenarios. To explore the personalized role-playing performance of small language models, we also conduct evaluation based on LLaMA-3.2-3B-Instruct and Qwen2.5-3B-Instruct. Separated results on small language models are shown in Table 2. The results shown in Table 1 demonstrate that the performance gap between small models and larger models in personalized role-playing task is not significant and under specific methods, small models even perform better than larger models. Specifically, for LLaMA-3.2-3B-Instruct, BERTScore and ROUGE scores are close to those of LLaMA-3-8B-Instruct, and the mean gap of Avg scores in LLM-as-Judge compared with LLaMA-3-8B-Instruct is 0.234. For Qwen2.5-3B-Instruct, we can observe that BERTScore and ROUGE scores surpass those of Qwen2.5-7B-Instruct, and under zero-shot setting, Qwen2.5-3B-Instruct outperform Qwen2.5-7B-Instruct across all LLM-as-Judge metrics, while the average gap of Avg scores in LLM-as-Judge under the other settings is 0.154. The surpass may be because Qwen2.5-3B-Instruct tend to generate concise responses which align more closely with the characteristics of social media comments. These findings highlight the value of developing methods for personalized adaptation on small language models.

### 5.3.3 CHALLENGE ANALYSIS: LENGTH OF TARGET COMMENTS

In this section, we investigate the relation between personalized role-playing performance and sequence length of ground-truth comments under RAG and SFT settings. Specifically, we partition the test samples of all users into bins based on the sequence length of ground-truth comments, and then compute the average score within each bin. The result is shown in Figure 2. We can observe that under the RAG setting, personalization scores exhibit a trend of initially rising then declining as target comment length increases, illustrating that for users accustomed to posting concise comments, the model may struggle to extract effective personalization information from historical interactions, thereby hindering personalization modeling. Conversely, for users accustomed to posting long comments, the personalization information provided by RAG proves insufficient to support long-context

Table 2: The average results across all target users for each evaluated role-playing method on LLaMA-3.2-3B-Instruct and Qwen2.5-3B-Instruct. In each group characterized by the original LLM, **bold** figures indicate the best result in each evaluation dimension, while the underlined figures denote the second best one.

| Method | BERTScore | ROUGE | | | LLM-as-Judge | | | |
|--------|-----------|-------|-------|-------|-----|-----|-----|-----|
| | | ROUGE-1 | ROUGE-2 | ROUGE-L | SS | CI | LS | Avg. |
| LLaMA-3.2-3B-Instruct | | | | | | | | |
| few-shot | 0.836 | 0.124 | 0.011 | 0.091 | 1.404 | 1.998 | 1.555 | 1.652 |
| RAG | 0.832 | 0.143 | **0.018** | **0.099** | 1.529 | 2.112 | 1.640 | 1.760 |
| SFT | **0.837** | 0.117 | 0.013 | 0.088 | 1.309 | 1.828 | 1.441 | 1.526 |
| DPO | 0.833 | 0.095 | 0.010 | 0.075 | 1.310 | 1.825 | 1.410 | 1.515 |
| GRPO | 0.825 | **0.157** | **0.018** | 0.096 | **1.925** | **2.814** | **1.998** | **2.246** |
| Qwen2.5-3B-Instruct | | | | | | | | |
| few-shot | 0.834 | 0.132 | 0.012 | 0.088 | 1.821 | 2.672 | 1.875 | 2.123 |
| RAG | 0.833 | 0.147 | **0.020** | **0.097** | 1.840 | 2.637 | 1.859 | 2.112 |
| SFT | **0.838** | 0.118 | 0.014 | 0.088 | 1.434 | 2.044 | 1.579 | 1.686 |
| DPO | 0.837 | 0.112 | 0.011 | 0.082 | 1.542 | 2.258 | 1.651 | 1.817 |
| GRPO | 0.828 | **0.153** | 0.016 | 0.092 | **1.980** | **2.963** | **2.013** | **2.319** |

personalized generation. Meanwhile, under the SFT setting, personalization scores exhibit a similar trend to RAG, which demonstrates that SFT also struggles to tackle the personalization modeling challenges posed by user interactions that are either excessively brief or excessively long. The results highlight the need to develop methods for mining, refining, and efficiently utilizing personalized information.

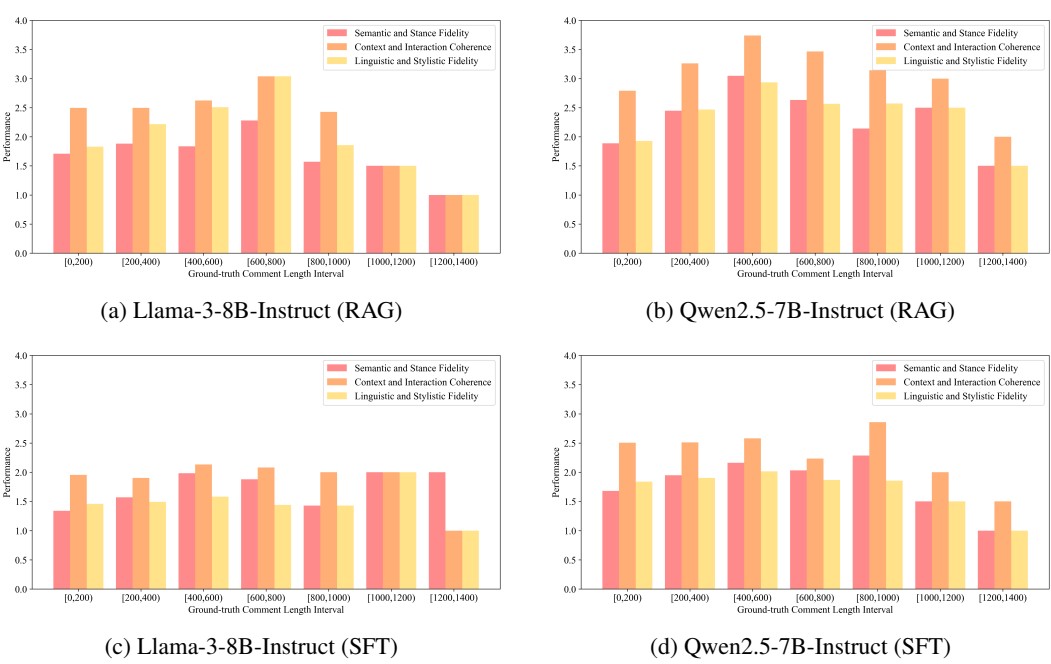

(a) Llama-3-8B-Instruct (RAG)

(b) Qwen2.5-7B-Instruct (RAG)

(c) Llama-3-8B-Instruct (SFT)

(d) Qwen2.5-7B-Instruct (SFT)

Figure 2: Average LLM-as-Judge scores of length bins under RAG and SFT settings.

### 5.3.4 CHALLENGE ANALYSIS: NOISE IN CONTEXT

Noise is ubiquitous on social media. In this section, we focus on the persuasive noise in the context of target comments created by malicious attackers posing as genuine participants in topic discussion. Such persuasive noise attempts to reverse the original polarity of target comments, thereby

Table 3: The attack success rate towards training-based methods including SFT and DPO.

| Method | $N_1$ | $N_2$ | $N_2/N_1(\%)$ |
|---|---|---|---|
| Llama-3-8B-Instruct (SFT) | 2,073 | 777 | 37.48 |
| Qwen2.5-7B-Instruct (SFT) | 1,931 | 997 | 51.63 |
| Llama-3-8B-Instruct (DPO) | 2,179 | 653 | 29.97 |
| Qwen2.5-7B-Instruct (DPO) | 1,665 | 959 | 57.60 |

compromising the fidelity of personalized role-playing approaches. The process of attack can be formalized as follows:

$$\hat{c_0}' = G(P, H') \tag{4}$$

$$H' = [(u_1, c_1), ...(u_N, c_N), (u'_{N+1}, c'_{N+1})] \tag{5}$$

where the noise $c'_{N+1}$ in the disturbed comment history $H'$ is introduced by the attacker $u'_{N+1}$. We define three polarities for the ground-truth comment $c_0$, the normal simulated comment $\hat{c}_0$, and the attacked simulated comment $\hat{c}_0'$: positive, neutral, and negative. We consider an attack successful when the polarity of $\hat{c}_0$ matches that of $c_0$, while the polarity of $\hat{c}_0'$ differs from that of $c_0$:

$$ASR = \frac{N_2}{N_1} \tag{6}$$

$$N_1 = |\{(c_0^{(j)}, \hat{c}_0^{(j)}, \hat{c}_0'^{(j)})|plr(\hat{c}_0^{(j)}) = plr(c_0^{(j)}), j \in \{1, 2, ..., M\}\}| \tag{7}$$

$$N_2 = |\{(c_0^{(j)}, \hat{c}_0^{(j)}, \hat{c}_0'^{(j)})|plr(\hat{c}_0^{(j)}) \neq plr(\hat{c}_0'^{(j)}), j \in \{1, 2, ...M\}\}| \tag{8}$$

where $j$ is the sample index and $M$ is the total number of samples, and $plr$ represents the polarity of the comment.

To explore the robustness of role-playing methods to the noise, we conduct an adversarial evaluation experiment, where GPT-4.1 is utilized as the attack model to analyze the views in the ground-truth comments and generate the persuasive noise. We show the attack results towards SFT-based and DPO-based personalized role-playing in Table 3. We can observe that all tested methods exhibit vulnerability to persuasion injected into context, with the lowest attack success rate reaching 29.97%. Meanwhile, the finetuned Qwen2.5-7B-Instruct is more susceptible to persuasive noise than the finetuned Llama-3-8B-Instruct. The results call for more research efforts into secure and controllable personalized role-playing.

# 6 CONCLUSION

In this paper, we propose the task of personalized role-playing on social media by reconstructing discussions under specific topics into utterance-level comment simulations to assist more fine-grained individual simulation. Considering the lack of readily available public datasets for task evaluation and the potential issue of data contamination, we construct a user-centered and newly collected dataset for evaluating existing role-playing methods. To comprehensively evaluate the performance of personalized role-playing, we introduce both classic text similarity metrics and LLM-as-Judge metrics across three key dimensions, referring to the comparison between the simulated comments and ground-truth comments. We classify existing role-playing methods into prompt-based and training-based categories and evaluate their performance on personalization. Evaluation results reveal that RAG and GRPO are the two methods that perform relatively better among the two categories. However, the generally low personalization scores indicate that existing role-playing methods struggle to achieve realistic personalized modeling. We also observe that merely scaling model parameters or utilizing reasoning models yields limited gains in personalization. The adversarial attack experiment uncovers the vulnerability of evaluated methods to persuasive noise injected into context. In future work, we will dedicate more efforts to developing role-playing methods capable of mining and efficiently utilizing personalized information from social media users, while simultaneously striving to enhance their security and robustness.

## 7 ETHICS STATEMENT

The data used in this study was collected from publicly available sources. We acknowledge that the dataset may contain societal biases. We release our code and model with a responsible AI license.

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

## A APPENDIX

### A.1 USE OF LLMS

We only use LLMs to refine the textual sections of the paper and selectively adopt the optimization suggestions provided by LLMs.

## A.2 IMPLEMENTATION DETAILS

In our evaluation, the loading and inference of open-source LLMs are implemented based on the HuggingFace's Transformers Library. Under RAG setting, the construction of index and retrieval of relevant interactions are implemented with faiss library using bge-m3 embedding model and inner-product similarity. The LoRA training for SFT and DPO is performed using LLaMA-Factory framework. Training and evaluation within GRPO are implemented with verl and vllm. For SFT and DPO, we set the number of training epoch to 3, with learning rate set to 1e-4, LoRA rank set to 8, and LoRA alpha set to 16. Experiments of SFT and DPO are conducted on 8 NVIDIA RTX 3090 GPUs. We enable length truncation during the training process, with the maximum input sequence length set to 2048 for SFT and 768 for DPO. For GRPO, we set the number of training epoch to 1 considering the time cost, with learning rate set to 1e-6, maximum prompt length set to 512, and maximum response length set to 1024. Experiments of GRPO are conducted on 4 NVIDIA A100 GPUs. We do not observe model collapse during the training of the above methods.

## A.3 PROMPT TEMPLATES

Assume you are a social media user. Write a highly realistic comment based on the content of the post and the existing comment thread.\n\n
**CRITICAL INSTRUCTIONS:**
- Analyze the context from the post title, body, and existing dialogue.
- Your entire response MUST BE the text of the comment itself.
- DO NOT include any prefixes(e.g. "Here is the comment:"), introductions, explanations, or quotation marks(i.e. "").\n\n
**Example**:
Post Title:{post_title_example}\n\n
Post Body:{post_body_example}\n\n
Dialogue History:
{dialogue_history_example}\n\n
Generated Comment:
{ground_truth_comment_example}\n\n
**Your Task**:
Post Title:{post_title_test}\n\n
Post Body:{post_body_test}\n\n
Dialogue History:\n{dialogue_history_test}\n\n
Generated Comment:\n

Figure 3: The prompt used to generate simulated comments under few-shot setting

Assume you are a social media user. Write a highly realistic comment based on the content of the post and the existing comment thread.\n\n
**CRITICAL INSTRUCTIONS:**
- Analyze the context from the post title, body, and existing dialogue.
- Your entire response MUST BE the text of the comment itself.
- DO NOT include any prefixes(e.g. "Here is the comment:"), introductions, explanations, or quotation marks(i.e. "").\n\n
**Relevant Comments You Made**:
Post Title:{post_title_retrieved}\n\n
Post Body:{post_body_retrieved}\n\n
Dialogue History:
{dialogue_history_retrieved}\n\n
Generated Comment:
{ground_truth_comment_retrieved}\n\n
**Your Task**:
Post Title:{post_title_test}\n\n
Post Body:{post_body_test}\n\n
Dialogue History:\n{dialogue_history_test}\n\n
Generated Comment:\n

Figure 4: The prompt used to generate simulated comments under RAG setting

You are a judge responsible for scoring a simulated social media comment given the topic post, comment thread and ground truth user comment.
The evaluation dimensions include Semantic & Stance Fidelity, Contextual & Interactional Coherence and Linguistic & Stylistic Fidelity.
The core evaluation point of Semantic & Stance Fidelity is how accurately the simulated comment reproduces the core idea, sentiment, and stance of the real comment.
The core evaluation point of Contextual & Interactional Coherence is how logical and interactive the simulated comment is within the context.
The core evaluation point of Linguistic & Stylistic Fidelity is how similar the word choice, tone, emoji use, and sentence structure of the simulated comment is to the real comment.
The evaluation utilizes a 5-point Likert scale with 5 representing strongly agree, i.e. the highest simulation quality.
Here are examples for each dimension:\n\n
**Semantic & Stance Fidelity**\n
Post Title: My Verdict on the New Orbit X1: A Battery Champ with One Major Flaw\n
Post Body: I've spent the last week with the Orbit X1, and the battery life is a game-changer. I'm consistently getting two full days of use, which is incredible. However, I can't ignore the screen's color calibration. It's far too saturated, and colors look almost cartoonish, making it a poor choice for anyone who does photo editing on their phone.\n
Comment Thread: \n
User A: Just finished watching your video, this generation feels like a huge improvement!\n
Ground Truth Comment: I agree with you, the battery is top-notch, but I'm personally not a fan of the screen color either, it's too vivid.\n
Positive Example (Rating: 5/5): True, the battery life is really impressive. The screen's color profile is a bit too rich for my taste, though. Something to watch out for if you're serious about color accuracy.\n
Analysis: This comment correctly captures both the positive stance on the battery and the negative stance on the screen, perfectly matching the semantics and opinion of the real comment.\n
Negative Example (Rating: 1/5): The battery and the screen on this phone are both amazing, it's a perfect all-rounder!\n
Analysis: This comment misinterprets the explicit criticism of the screen, completely contradicting the real comment's stance. It fails the semantic fidelity test.\n\n
**Contextual & Interactional Coherence**\n
Post Title: I Tried to Make a 3-Day \"Buddha Jumps Over the Wall\" Soup At Home...\n
Post Body: This was by far the most ambitious recipe I've ever attempted. From sourcing two dozen ingredients (some of which were very hard to find!) to the multi-day prep and simmering process, it was an absolute marathon. The final result was incredible, but was it worth it? Check out the video to see the full journey!\n
Comment Thread: \n
User A: OMG, just prepping the ingredients must take two days, right? So impressive!\n
User B: I know, right? I got hungry just watching, but I would never have the guts to make it myself.\n
Ground Truth Comment: Don't even start, I don't even have the courage to buy the ingredients 😂\n
Positive Example (Rating: 5/5): Hahaha, same here. I basically gave up just by looking at the shopping list.\n
Analysis: This comment perfectly understands the conversational context. It logically follows User B's sentiment about the difficulty and responds with relevant humor, demonstrating strong interactional coherence.\n
Negative Example (Rating: 1/5): Buddha Jumps Over the Wall' is a variety of shark fin soup in Fujian cuisine, a specialty of Fuzhou, China.\n
Analysis: This comment completely ignores the conversation's flow about the effort of cooking. It drops a random, encyclopedic fact, failing to interact with the previous comments and breaking the context.\n\n
**Linguistic & Stylistic Fidelity**\n
Post Title: CYBERPUNK: VENGEANCE - OFFICIAL TRAILER 1\n
Post Body: The wait is over. Witness the first official look at CYBERPUNK: VENGEANCE. Coming to theaters this December. #CyberpunkVengeance #MovieTrailer\n
Comment Thread: \n
User A: OMG! That CGI! It's insane!!\n
User B: You can just feel the budget burninggggg, already booked my IMAX tickets!\n
Ground Truth Comment: That was so cool it gave me goosebumps!!! Watching this 3 times minimum, seriously!!! LETS GOOOO!!\n
Positive Example (Rating: 5/5): OMG I'm literally getting goosebumps it's so hype! LFG! Definitely rushing to see this on day one!!\n
Analysis: The style is a perfect match. It uses authentic internet slang (\"hype,\" \"LFG\"), captures the excited tone with punctuation, and mimics the sentence structure of a genuinely thrilled fan.\n
Negative Example (Rating: 1/5): Upon observation, the visual effects presented in this trailer are quite impressive. The fluid choreography and high-quality rendering are commendable. I recommend releasing it soon.\n
Analysis: This comment sounds like a formal report, not a fan. The stiff, academic language is completely out of place with the energetic and informal style of the real comments, failing the stylistic test.\n\n
Now here is the case to be evaluated:\n
Post Title: {post_title}\n
Post Body: {post_body}\n
Comment Thread: {dialogue_history}\n
Ground Truth Comment: {gt_reply}\n
Simulated Comment: {test_reply}\n
You are required to output evaluation in JSON format like below: \n
{{\"Semantic & Stance Fidelity\": {{\"score\":, \"analysis\":}}, \"Contextual & Interactional Coherence\": {{\"score\":, \"analysis\":}}, \"Linguistic & Stylistic Fidelity\": {{\"score\":, \"analysis\":}},"

Figure 5: The prompt used to evaluate simulated comments and provide rewards for GRPO training

