# OpenReview forum: "Imitation Game: Toward Comprehensive Evaluation on Personalized Role-Playing on Social Media"
_ICLR.cc/2026/Conference — ICLR 2026 Conference Withdrawn Submission_

### Official Review · Reviewer_Af9f · 2025-10-28

**Soundness:** 1
**Presentation:** 2
**Contribution:** 2
**Rating:** 4
**Confidence:** 3

**Summary:**

This paper operationalizes individual-level role-play on social media by formulating a conditional generation task (i.e. predicting the comment a specific user would write given the thread context on reddit). It constructed IDRole dataset and evaluated LLM over it via different prompting or fine-tuning methods. It also performs robustness analyses to context manipulation, revealing salient failure modes and current limitation in current personalization methods.

**Strengths:**

1. The paper reframes “write like this user” into a precise conditional generation task with a user-centric benchmark, extending personalization beyond fictional role-play to real social discourse.

2. Presentation, methodology, and results are clear: the task is well specified and generally easily to follow and understand.

**Weaknesses:**

1. It's doubtful of whether it's suitable to directly choose user-generated content without strong filters. User contents are actually noisy and might be impacted by many external factors (i.e. real life experiences). Personalization of LLM may not be poor per se; rather, without these additional signals and relying solely on text inference, the target might be underdetermined.

2. To support the conclusions, repeated trials should be reported with mean and standard deviation.

3. The comparisons include prompting, RAG, SFT, and PPO/GRPO, yet widely used personalization baselines (e.g., persona/author-style modeling [1]) are not presented, whose inclusion would strengthen the empirical studies.

[1] Zhang, Zhehao, et al. "Personalization of large language models: A survey." arXiv preprint arXiv:2411.00027 (2024).

**Questions:**

Please refer to the weakness.

**Details Of Ethics Concerns:**

Please verify that whether the reddit content is collected under license or permit. I’m not fully versed in Reddit’s internal rules, but they do have license requirement for data usage https://support.reddithelp.com/hc/en-us/articles/26410290525844-Public-Content-Policy

---

### Official Review · Reviewer_PJTK · 2025-10-30

**Soundness:** 3
**Presentation:** 3
**Contribution:** 2
**Rating:** 4
**Confidence:** 3

**Summary:**

This paper proposes a novel task called Personalized Role-playing on Social Media (PRISM)—aimed at addressing the lack of individual heterogeneity capture in existing social simulation by requiring LLMs to mimic ordinary users’ perspectives, stances, and linguistic habits for comment generation—and constructs the IDRole benchmark (67 users, 21K comments, and 7K posts). It evaluates four role-playing methods (few-shot prompting, SFT, DPO, GRPO) using classical metrics (BERTScore, ROUGE) and LLM-as-Judge metrics (covering Semantic & Stance Fidelity, Contextual & Interactional Coherence, Linguistic & Stylistic Fidelity) across open-source and commercial LLMs. Key findings include that existing methods struggle with fine-grained personalization, scaling model parameters or using reasoning models fails to boost personalization, evaluated methods are vulnerable to context noise, and small LLMs show potential, while the work’s main contributions lie in introducing PRISM, building IDRole, and offering insights for robust personalized role-playing methods.

**Strengths:**

1. Innovative Task Design: The paper proposes the PRISM task (Personalized Role-playing on Social Media), specifically targeting ordinary social media users, filling the research gap in fine-grained personalized role-playing for individuals.
2. Valuable Benchmark Construction: It builds the IDRole benchmark, addressing the scarcity and contamination of public social media datasets, with well-processed utterance-level samples and anti-leakage sorting (by comment depth).
3. Comprehensive Evaluation Dimension: The study combines classical similarity metrics (BERTScore, ROUGE) and LLM-as-Judge (covering SS/CI/LS), evaluates diverse methods (few-shot, SFT, DPO, GRPO), and explores small models/noise robustness—providing systematic insights.

**Weaknesses:**

1. Limited Dataset Generalizability: The IDRole dataset includes only 67 Reddit users and no data from other mainstream platforms (e.g., Twitter, Facebook), making its conclusions hard to extend to diverse social media users or platforms.
2. Incomplete Model Evaluation: Commercial LLMs are only represented by GPT-4.1 (excluding Claude/Gemini), and the "small model potential" analysis lacks depth (e.g., no topic/user-specific performance breakdown).
3. Unverified LLM-as-Judge Reliability: The study uses only GPT-4.1 as the judge, with no validation via other LLMs or human evaluations to confirm the accuracy of its SS/CI/LS scoring.

**Questions:**

1. Both BERTScore and ROUGE are used to measure response similarity, so why do they differ significantly?
2. What is the difference between Table 1 and Table 2?
3. How is the user profile obtained?
4. How do models trained on other role-playing datasets (eg, COSER[1] ) perform on this task?


[1] Wang, X., Wang, H., Zhang, Y., Yuan, X., Xu, R., Huang, J., Yuan, S., Guo, H., Chen, J., Wang, W., Xiao, Y., & Zhou, S. (2025). CoSER: Coordinating LLM-Based Persona Simulation of Established Roles. ArXiv, abs/2502.09082.

**Details Of Ethics Concerns:**

This article involves private data from social media.

---

### Official Review · Reviewer_1Tew · 2025-10-30

**Soundness:** 2
**Presentation:** 2
**Contribution:** 2
**Rating:** 4
**Confidence:** 4

**Summary:**

This paper introduces a new task called Personalized Role-Playing on Social Media (PRISM), aiming to evaluate how well LLMs can imitate ordinary users’ behaviors in social media discussions. To support this, the authors construct a new dataset IDRole, derived from Reddit, containing 67 users, 7K posts, and 21K comments. The task is evaluated using both classical similarity metrics (ROUGE, BERTScore) and LLM-as-Judge metrics across three personalization dimensions. The paper comprehensively benchmarks multiple role-playing approaches across various open-source and commercial models. The results reveal some findings and drawbacks.

**Strengths:**

1. Novel Task Definition: The proposed PRISM task fills an important gap between LLM-based social simulation and persona-based role-playing, shifting the focus from well-defined fictional or celebrity roles to ordinary social media users.
2. Comprehensive Evaluation of Methods: The authors test a wide spectrum of methods (prompting, fine-tuning, preference optimization, reinforcement learning).

**Weaknesses:**

1. Dataset Scale and Diversity: The dataset covers only 67 users, small for a “social simulation” study. It risks overfitting to narrow linguistic or topical communities (e.g., science/technology subreddits, or the most active users).
2. Evaluation Dependency on LLM-as-Judge: Relying on GPT-4.1 as the judge introduces bias. The paper would benefit from human validation.
3.  Limited Model Coverage: The evaluation omits stronger frontier models, such as OpenAI o1/o3, Gemini 2.5, Claude 4, Qwen 3, and DeepSeek R1, whose advanced reasoning and personalization abilities could provide valuable baselines. This limitation weakens the claim of a “comprehensive evaluation”.
4. Analysis Depth: Some analyses (e.g., length-based and noise-based) are interesting but remain descriptive. There is no diagnostic exploration of why certain models fail or what linguistic cues they miss in imitation.

**Questions:**

1. In Line 198, the ref to Figure 5 may be wrong.
2. Tables 1 and 2 redundantly report the performance of LLaMA-3.2-3B-Instruct, which is unnecessary.
3. What is the detailed implementation of LLM-as-Judge? How to get the specific score values in LLM-as-Judge?
4. What possible directions does the author think could be used to address the shortcomings of the existing methods mentioned in the paper?

**Details Of Ethics Concerns:**

The dataset is derived from posts and comments on Reddit and may contain personal privacy information.

---

### Official Review · Reviewer_coPN · 2025-10-31

**Soundness:** 2
**Presentation:** 2
**Contribution:** 1
**Rating:** 2
**Confidence:** 4

**Summary:**

This paper argues that current work in social simulation and role-playing mostly overlooks ordinary individuals, and instead introduces a personalized role-playing task grounded in social media interactions. The authors collect Reddit posts and comments via API to build the IDRole Benchmark. Using this benchmark, they compare a range of role-playing approaches (including few-shot prompting, SFT, DPO, and GRPO) and evaluate multiple model families. The performance is assessed through semantic and lexical similarity as well as LLM-as-Judge protocol. The results provide empirical insights into the strengths and limitations of these personalized role-playing methods.

**Strengths:**

1.	The paper is clearly written and easy to follow.
2.	It provides a thorough comparison of several mainstream role-playing approaches (few-shot prompting, SFT, DPO, and GRPO).
3.	The work includes additional analyses, such as the effect of target comment length and the impact of noise on performance.

**Weaknesses:**

1.	The contributions of this paper are rather limited. Regarding the first contribution, the proposed task is to simulate how users write comments, which is fundamentally a personalized writing task [1] and not particularly novel. As for the second contribution, collecting data via API is a minor effort, and the dataset itself is very small (only 67 users). This scale is hardly representative for a benchmark claiming to support personalized social simulation. Moreover, large-scale Reddit datasets are already publicly available [2].
2.	The experimental results are not very convincing. First, the chosen metrics (ROUGE and BERTScore) are of limited value in the LLM era. As shown in the paper, BERTScore exhibits minimal differentiation across models (highest 0.838 vs. lowest 0.825). In addition, relying solely on LLM-as-Judge is insufficient for a complete evaluation—it should be complemented with human assessment or further validation. The experimental setups also appear overly simplistic. For instance, the few-shot configuration uses only two historical comments, which may not meaningfully test the model’s role-playing ability. Furthermore, the finding that SFT and DPO underperform few-shot prompting contradicts existing literature on role-playing tasks. This raises questions about the task design and training implementation. Given the small dataset (only 67 users), the model should have seen all users during SFT and thus exhibit stronger imitation capability.
3.	Several writing issues:
  - The RAG method is not described in the paper but still appears in the results table.
  - Table 2 repeats the content of Table 1.
  - Line 163 uses inconsistent subscripts for uₒ and c₀; moreover, the paper does not clarify how uₒ’s profile is incorporated. Without profiles, the claim of personalization is unconvincing.
  - In Section 2.2, the title should use “LLM-based” instead of “LLMS-based.”


[1] Kumar I, Viswanathan S, Yerra S, et al. Longlamp: A benchmark for personalized long-form text generation[J]. arXiv preprint arXiv:2407.11016, 2024.

[2] Baumgartner J, Zannettou S, Keegan B, et al. The pushshift reddit dataset[C]//Proceedings of the international AAAI conference on web and social media. 2020, 14: 830-839.

**Questions:**

Refer to weaknesses

---

### Note · Authors · 2026-01-05

I have read and agree with the venue's withdrawal policy on behalf of myself and my co-authors.